# Insights into Early Ontogenesis of *Salmo salar*: RNA Extraction, Housekeeping Gene Validation and Transcriptional Expression of Important Primordial Germ Cell and Sex-Determination Genes

**DOI:** 10.3390/ani13061094

**Published:** 2023-03-19

**Authors:** Irfan Ahmad Bhat, Milena Malgorzata Dubiel, Eduardo Rodriguez, Zophonías Oddur Jónsson

**Affiliations:** 1Institute of Life and Environmental Sciences, School of Engineering and Natural Sciences, University of Iceland, 101 Reykjavik, Iceland; 2Benchmark Genetics, 220 Hafnarfjordur, Iceland

**Keywords:** PGCs, gonads, sex determination, housekeeping genes, *S. salar*, sterility

## Abstract

**Simple Summary:**

The development of primordial germ cells (PGCs) and sex determination (SD) is governed by a complex interplay of genes that can be targeted to hinder sexual development in fish. Obtaining high-quality ribonucleic acid (RNA) from Atlantic salmon embryos, especially after fertilization, can be extremely challenging due to the presence of a substantial amount of yolk. The objective of this research was to extract high-quality RNA from developing salmon embryos for use in downstream applications. The study also aimed to validate different housekeeping genes (HKGs) for a quantitative polymerase chain reaction (qPCR)-based assessment of PGC and SD genes during developmental stages in salmon. We isolated RNA from the developing embryos, and we present the validation of HKGs as well as the mRNA expression levels of important PGC and SD genes during the fertilization to hatching stage. The findings of this study could prove beneficial for researchers seeking to extract high-quality RNA from salmonids. Moreover, the transcript results may offer insights not only into the function of transcripts at specific developmental stages but also in identifying the stage with the highest expression levels, during which treatment to disrupt the function of the transcripts and ultimately the growth of PGCs could be administered.

**Abstract:**

The challenge in extracting high-quality RNA impedes the investigation of the transcriptome of developing salmonid embryos. Furthermore, the mRNA expression pattern of important PGC and SD genes during the initial embryonic development of *Salmo salar* is yet to be studied. So, in the present study, we aimed to isolate high-quality RNA from eggs and developing embryos to check *vasa*, *dnd1*, *nanos3a*, *sdf1*, *gsdf*, *amh*, *cyp19a*, *dmrt1* and *foxl2* expression by qPCR. Additionally, four HKGs (*GAPDH*, *UB2L3*, *eEf1a* and *β-actin)* were validated to select the best internal control for qPCR. High-quality RNA was extracted, which was confirmed by spectrophotometer, agarose gel electrophoresis and Agilent TapeStation analysis. *UB2L3* was chosen as a reference gene because it exhibited lower intra- and inter-sample variation. *vasa* transcripts were expressed in all the developmental stages, while *dnd1* was expressed only up to 40 d°C. *Nanos3a* was expressed in later stages and remained at its peak for a shorter period, while *sdf1* showed an irregular pattern of mRNA expression. The mRNA expression levels of SD genes were observed to be upregulated during the later stages of development, prior to hatching. This study presents a straightforward methodology for isolating high-quality RNA from salmon eggs, and the resulting transcript profiles of significant PGC and SD genes in *S. salar* could aid in improving our comprehension of reproductive development in this commercially important species.

## 1. Introduction

Atlantic salmon (Salmo salar of the Salmonidae family) is an anadromous fish that is native to rivers flowing into the North Atlantic Ocean from Northern Europe as far south as Spain, the Baltics, Canada and Northeastern USA [1]. Many natural Atlantic salmon populations have declined significantly due to overexploitation [2] and other anthropogenic factors [3]. Due to its popularity as food and overexploitation in the wild, there has been a considerable shift towards the farming of the species under controlled conditions. Ocean pen and land-based farming of A. salmon has grown into a billion-dollar industry [4,5], and they are currently experiencing exponential growth. However, the farming of salmon can have negative impacts on wild populations, as the farmed fish exhibit different breeding and feeding behavior and aggressive tendencies, while also potentially transmitting harmful diseases to wild populations [6,7,8].

Different policies, guidelines and procedures have been formulated to prevent the escape of farmed salmon [9,10]. Physical and environmental containment is not secure enough to prevent fish from escaping into the wild [11], which necessities focusing more on biological containment. In biological containment, sexual reproduction is disrupted so that the fish produced is completely sterile, and this is the best safeguard against the spread of cultured stocks by eliminating their reproductive interaction with wild fish [12]. Moreover, the sterile population also has better production time, flesh quality and growth by eliminating energy costs associated with breeding [13,14]. Triploid fish are sterile, but at the same time, their disease resistance and tolerance to the changing environment is low [15,16] making them sub-optimal for aquaculture. Disruption of the growth of primordial germ cells (PGC), which are required to produce germ cells in both sexes during embryonic stages, is becoming a popular way to achieve sterility in fish [17,18,19]. The most effective approach to prevent the formation of PGCs is to silence the genes responsible for their growth and migration. [20]. Surprisingly, the mRNA expression profile of most of the genes involved in the development of PGCs in A. salmon during the initial embryonic stages is yet to be studied in detail. Gaining knowledge of the mRNA expression levels of these genes can aid in tracking the effectiveness of treatments used to induce sterility by eliminating PGCs during the embryonic stage. The bottleneck for the analysis of transcript levels of different genes during embryonic stages is the extraction of quality mRNA from the large yolk-filled eggs. The RNA extraction from cyprinid eggs [mostly used as model organisms] is quite different from other fish species such as salmonids because of the larger egg diameter of the latter [21]. The larger egg diameter coincides with higher yolk content, making it difficult to isolate pure RNA from them. Additionally, due to the low cell-to-lipid ratio in fish eggs, RNA isolation from eggs is challenging, especially when it needs to be used for downstream applications [22]. In molecular biological and diagnostic applications, it is preferable to use high-quality RNA as a starting point. The quantity and quality of starting RNA are recognized to have an impact on the accuracy of gene expression evaluation. The difficulty in isolating pure RNA hinders the study of transcriptomic changes in unfertilized eggs and during the initial embryonic development of fishes, especially salmonids.

The most important step for the analysis of quantitative polymerase chain reaction (qPCR) data is normalization with the proper housekeeping gene (HKG) to eliminate pre-PCR and PCR processing variations [23,24]. The choice of an appropriate HKG is crucial for accurate and reliable results in qPCR analysis, as any variability or bias in their expression can affect the normalization and interpretation of the target gene expression. An ideal reference gene should be constantly expressed in the testing samples to prevent erroneous results [25]. Often, however, HKGs under different experimental conditions or developmental stages show variable results [26,27]. It is advised to validate more than one HKG for the normalization of qPCR data to avoid misleading results [28,29]. By validating housekeeping genes, we can ensure the reliability and reproducibility of qPCR experiments and minimize the risk of misinterpreting gene expression data.

In the present study, we developed an efficient way to extract cells from the developing embryo for RNA extraction. Additionally, the expression of four reference genes—glyceraldehyde 3-phosphate dehydrogenase (*GAPDH*), ubiquitin-conjugating enzyme E2 L3 (*UB2L3*), elongation factor 1-alpha (*eEf1a*) and beta-actin (*β-actin*)—to select the best internal control was assessed by qPCR. The stability in mRNA expression of the reference genes was validated using different statistical algorithms.

In addition, we examined the mRNA expression levels of several key genes, including *vasa*, dead-end gene (*dnd1*), *nanos3a* and stromal cell-derived factor 1 (*sdf1*), which are involved in the growth, development and migration of PGCs [30]. *vasa* belongs to the ATP-dependent RNA helicase of the DEAD (Asp–Glu–Ala–Asp)-box family and performs roles in germ cell origination, migration and development [31,32]. The *dnd* gene encodes an RNA-binding protein that is essential for PGC migration and prevents the degradation of germplasm RNAs by binding with microRNA that targets it [33,34]. *nanos3a* plays an important role in the migration and maintenance of germ cells [35,36,37,38]. *sdf1* (chemokine ligand 1) is secreted by the somatic cells surrounding PGCs and provides chemoattractant signals, which are sensed by its receptor CXCR4B expressed on the PGCs itself and help in their migration [39]. Knockdown studies of *sdf1* genes in fish disrupted the migration of PGC [40].

Similarly, the transcript profile of some important transcription factors required in SD and gonadal development in fish were analyzed during embryogenesis in A. salmon. The genes included gonadal somatic derived factor (*gsdf*), doublesex and mab-3 related transcription factor (*dmrt1*)), anti-Mullerian hormone (*amh*), forkhead box protein L2 (*foxl2*) and cytochrome P450arom (*cyp19a*) [41,42,43].

## 2. Materials and Methods

### 2.1. Fertilization and Embryo Development of S. salar

Fertilization of salmon eggs was carried out in the Vogar fish farm of Benchmark Genetics, Iceland, following a standard operating protocol of the company. Eggs were collected from mature *S. salar* by an abdominal cavity excision after being anesthetized with 2- phenoxyethanol (at 0.1 to 0.6 mL∙L^−1^). An equal volume of eggs from four different females along with the ovarian fluid were obtained and kept in the same container and mixed with cryopreserved milt for fertilization. At three hours post-fertilization, the eggs were transported to the wet lab of the Institute for Experimental Pathology, University of Iceland, Keldur. The fertilized eggs were subsequently moved to glass tanks (capacity 50 L) to undergo development. The water temperature (10 °C), water flow and dissolved oxygen were maintained throughout the experiment. A water temperature of 10 °C is considered optimum for the development of *S. salar* eggs [44]. Dead eggs (whitish) were constantly removed from the tanks to maintain the water quality and to prevent the growth of any fungal infection.

### 2.2. Sampling

The unfertilized eggs at day one and fertilized eggs during developmental stages (fertilization to hatching) (Table 1) were collected for RNA extraction. Ten eggs/embryos until 40 degree days (d°C), six embryos up to 340 d°C and three yolk sac fry or alevins (only the trunk portion was collected) were collected and pooled together, and four such pools were used to study gene expression. The detailed sampling is presented in Table 1.

### 2.3. Cell Extraction from the Embryo during Different Developmental Stages

After each sampling, the developing eggs were treated with 5% acetic acid for 5 min (to make the chorion transparent). The chorion was then carefully removed, and the cell mass was separated from the yolk using a pipette. The collected cells were placed into a 1.5 mL tube with 500 µL of ice-cold PBS. The cells were centrifuged at 12,000× *g* for 7 min, and the supernatant was removed without disturbing the cells. The step was repeated two times to remove any remaining yolk. The unfertilized eggs were immersed in water for 1–3 h and were then placed into acetic acid to remove the cell by dissection.

### 2.4. RNA Extraction and cDNA Synthesis

Total RNA from the embryonic cells was extracted using the TRI reagent (Sigma-Aldrich, St. Louis, MO, USA) extraction method. The samples were homogenized (Mini-Beadbeater, Sigma-Aldrich) using stainless steel beads of 3 mm diameter, and following the phase separation, precipitation and washing steps, the pellet obtained was dissolved in water. RNA integrity and yield were confirmed by NanoDrop™ 2000/2000c Spectrophotometer (ThermoFisher Scientific, Waltham, MA, USA) gel electrophoresis, TapeStation (Agilent 2200 TapeStation system, Agilent, St. Clara, CA, USA) and nanodrop spectrophotometer (Thermofisher Scientific), respectively. The RNA samples were prepared as per the protocol suggested by Agilent RNA ScreenTape System Quick Guide, run on 2200 TapeStation instrument and analyzed with 2200 TapeStation Controller Software. RNA was treated with *DNase 1* to remove any DNA contamination. RNA was transcribed to complementary deoxyribonucleic acid (cDNA) using ProtoScript^®^ II First Strand cDNA Synthesis Kit (New England Biolabs Inc., Ipswich, MA, USA) by oligo dT priming following the manufacturer’s instructions. The cDNA samples were stored at −20 °C prior to analysis.

### 2.5. Primers

All the primers except for *dnd1* used in the present study were designed by Gene Runner software version 6.5.52 Beta. The mRNA sequences of all the genes of *S. salar* were retrieved from GenBank, NCBI, and the primer set was designed from the coding (cds) region. The dimers, hairpin loops, bulge loops, internal loops and match site were checked by the oligo analysis option of Gene Runner software. The primers were supplied by Microsynth AG company (Balgach, Switzerland) in a lyophilized form, which was dissolved in water and stored at −20 °C for further use. The primers were used to amplify the cDNA from *S. salar* embryos in RT-PCR, and the product was run on a 2% agarose gel. Only one band could be found at an expected size, which allowed us to use them in qPCR. In addition, a melting curve analysis was also performed to check the presence of only a single peak. The correlation coefficient (R^2^) of the primers was determined by serial dilution (10-fold) to generate the standard curve, and a value above 0.98 was accepted. The primer set for *dnd1* was taken from the previous report [19]. The primer sequences used in the present study are presented in Table 2.

### 2.6. Quantitative PCR (qPCR) Analysis

Real-time PCR was performed in 96-well PCR plates on a QuantStudio Real-Time PCR Systems (Thermo Fisher Scientific) using SYBR green, Luna^®^ Universal qPCR Master Mix (New England Biolabs Inc.) following manufacturers protocol with a slight modification in volume. Each sample was run in duplicate with a final volume of 10 µL reaction mixture per well. For each gene, a non-template control was also included in duplicate with 10 µL in volume. The qPCR program started with a 2 min hold at 50 °C followed by a 10 min hot start at 95 °C. It was followed by 40 cycles at 95 °C for 15 s and then the annealing temperature optimized for each primer set for 30 s. The melt curve analysis confirmed the presence of no dimer in the primers. The primer and PCR efficiency was also calculated from the slope of a standard curve generated by serials dilution of primers and cDNA. The expression analysis of the genes during different developmental stages was performed by using the 2^−ΔCt^ method [45] using the *UB2L3* gene as an internal control.

### 2.7. Validation of HKGs

The validation of the four HKGs, *GAPDH*, *UB2L3*, *eEf1a* and *β-actin,* was conducted to select the best internal control for the present study.

### 2.8. Data Analysis

Statistical analysis for the difference in expression levels of genes was carried out by a one-way ANOVA using SPSS 22.0 software (SPSS Inc., Chicago, IL, USA). Significant differences in mRNA expression of the genes were tested by one-way ANOVA, followed by Tukey’s HSD test (*p* < 0.05). Data are presented as mean ± standard deviation (SD), and *p* < 0.05 was considered statistically significant. To determine the stability of selected reference genes, different statistical algorithms were employed to analyze the set of data, i.e., the comparative delta Ct method [46], BestKeeper (version 1) [47], NormFinder (version 0.953) [48] and geNorm [49]. To select the most suitable reference gene, the results from the four methods were combined and screened by RefFinder [http://150.216.56.64/referencegene.phpm accessed on 20 September 2022], which gives comprehensive stability of HKGs and ranks them as per the stability [50].

## 3. Results

### 3.1. Embryonic Development

The fertilized eggs were allowed to develop at 10 °C, and their development was constantly monitored under the microscope. The embryos did not show any abnormalities that could vary the mRNA expression results.

### 3.2. RNA Quality

The nanodrop spectrophotometer showed a 260/280 ratio close to 2.0. The RNA in gel electrophoresis showed well-defined bands of 28s rRNA and 18s rRNA representing good-quality RNA, and TapStation results presented an RNA integrity number (RIN) value of more than eight. The figures of gel electrophoresis and the TapStation system are presented in Figure 1.

### 3.3. Validation of HKG

#### 3.3.1. Expression Level of Selected Candidate Reference Genes by Cycle Threshold (Ct) Value Analysis

The Ct values obtained from the qPCR assay across various samples were utilized to assess the expression levels and variability of all potential reference genes. *eEf1a* was expressed the most (Ct values 18 to 25), followed by *UB2L3* (Ct values 20 to 24), *β-actin* (Ct values 21–25) and *GAPDH* (Ct values 24–28). *UB2L3* exhibited the least variation of Ct values in different samples than other HKGs (Figure 2).

#### 3.3.2. Gene Stability Analysis of HKGs

The use of Ct values alone may not adequately indicate the stability of reference genes, highlighting the need to employ diverse statistical algorithms for data analysis. These algorithms can help determine the most robust reference genes that exhibit consistent expression levels across various experimental conditions and sample types. The stability analysis, which is based on the values generated from the delta Ct method, BestKeeper, NormFinder and geNorm algorithms, was used to generate the compressive stability (RefFinder) by which the best internal control was selected.

The analysis of the comparative delta Ct method is based on the average standard deviation, with a greater value indicating less stability and vice versa. *UB2L3* showed the lowest average standard deviation (1.64), followed by *GAPDH* (1.96), *β-actin* (2.09) and *eEf1a* (2.30). This method indicates *UB2L3* as the most stable HSK and *eEf1a* least stable.

The BestKeeper algorithm calculates the standard deviation of the data to predict the stability. The gene that shows the lowest value of standard deviation is considered the most stable internal control. In the present study, the lowest value was calculated for *UB2L3* (0.89), and the highest value was calculated for *eEf1a* (1.64), indicating the former as the more stable candidate reference gene. For *GAPDH,* the calculated value was 1.23, and for *β-actin,* it was 1.28. The statistical analysis of HKGs by the BestKeeper algorithm is presented in Table 3. So according to the BestKeeper algorithm, the order of ranking of the HKGs was *UB2L3*, *GAPDH*, *β-actin* and *eEf1a*.

The NormFinder algorithm calculates the stability value of each HKG and ranks them accordingly, i.e., a gene with the lowest value is the most stable and vice versa. Based on this analysis, the decreasing order of values calculated for different HSK genes was 0.65 > 1.42 > 1.57 > 1.97 for *UB2L3*, *GAPDH*, *β-actin* and *eEf1a,* respectively. The data from the NormFinder algorithm indicated *UB2L3* as the most stable and *eEf1a* as the least stable candidate reference genes.

Similarly, geNorm calculates the stability value and ranks the reference genes accordingly. The lowest value indicates more stability, and, in our case, *UB2L3* and *GAPDH* showed a similar lowest value (1.291), indicating that they were more stable candidate reference genes than *β-actin* (1.698) and *eEf1a* (1.997). The stability analysis calculated by the delta CT method, BestKeeper, NormFinder and geNorm algorithms is presented in Figure 3.

The average stability of the three HKGs was finally elucidated by the comprehensive gene stability calculated from the RefFinder, with the lower values representing the most stable genes (Figure 4). Here, *UB2L3* was reported to be the most stable candidate gene (average stability value of 1.0), followed by *GAPDH* (1.68), *β-actin* (3) and *eEf1a* (4).

Thus, from the above results, *UB2L3* was the most stable HKG and was selected as an internal control to study the mRNA expression of different PGC and SD genes during the ontogenetic development of *S. salar*.

### 3.4. mRNA Expression of Genes Involved in PGC Development of S. salar Collected from Fertilization to Hatching Stages

#### 3.4.1. *vasa*

The *vasa* transcript was found to be expressed significantly (*p* < 0.05) more in the embryos collected at 20 and 30 d°C. Up to 50 d°C, *vasa* mRNA expression was observed to be higher compared to unfertilized eggs. The mRNA expression was observed to be downregulated between 40 °C to 340 d°C, followed by an increase in expression at 430 d°C.

#### 3.4.2. *dnd1*

The expression of *dnd1* transcripts was found to be significantly higher (*p* < 0.05) in developed embryos collected at 10 d°C compared to unfertilized eggs. However, there was no significant difference (*p* < 0.05) in the transcript level between samples collected at 20 d°C and 30 d°C. The transcript was downregulated after 40 d°C and was almost absent in the later stages analyzed in this study.

#### 3.4.3. *nanos3a*

The expression of *nanos3a* transcripts was found to be significantly (*p* < 0.05) upregulated at 40 d°C and 50 d°C. A moderate level of expression was also observed at 30 d°C, 70 d°C and 80 d°C. However, a decreasing trend in *nanos3a* expression was observed from 110 d°C onwards, reaching levels similar to those observed in unfertilized eggs and the initial two stages analyzed in this study.

#### 3.4.4. *sdf1*

The expression pattern of the *sdf1* transcript was irregular, with significantly higher expression observed at 10 d°C, followed by 30 d°C and 20 d°C. The expression level was drastically downregulated up to 210 d°C, followed by an upward trend in the subsequent stages analyzed in this study.

The graphical representation of different PGC genes during the developmental stages of *S. salar* is presented in Figure 5.

### 3.5. Expression of SD Genes in S. salar during Embryonic Developmental Stages

#### 3.5.1. *gsdf*

The expression of this transcript was not detected until the late stages, beginning at 270 d°C. Significantly higher (*p* < 0.05) expression was observed in the hatching stage.

#### 3.5.2. *amh*

The expression of *amh* was not detected until 210 d°C, after which it was expressed in the subsequent stages analyzed in this study. Significantly higher (*p* < 0.05) expression was observed at 430 d°C.

#### 3.5.3. *cyp19a*

The expression pattern of *cyp19a* mRNA was similar to that of *amh*, with significantly higher (*p* < 0.05) expression observed at 430 d°C. No expression was detected in the unfertilized egg, and very low mRNA expression was detected up to 210 d°C.

#### 3.5.4. *dmrt1*

The expression of *dmrt1* showed a non-significant (*p* < 0.05) mRNA expression trend from the unfertilized egg to 270 d°C. However, in the 340 d°C and 430 d°C stages, the expression of *dmrt1* was significantly increased compared to the other stages analyzed in this study.

#### 3.5.5. *foxl2*

The expression of *foxl2* mRNA was very low in unfertilized eggs and developing embryos up to 80 d°C. However, an increasing trend in mRNA expression was observed from 110 d°C onwards, with significantly higher expression observed at 430 d°C.

The graphical representation of different SD/sex-differentiation genes during the developmental stages of *S. salar* is presented in Figure 6.

## 4. Discussion

Gonadal SD is a significant event that occurs during embryogenesis to form a bipotential gonad, and its fate to turn into testes or ovaries is determined by different genetic or environmental cues [51,52]. Similarly, gametes originate from specialized cells known as PGCs that are formed in the early stages of embryonic development and migrate to the developing gonads under the influence of different molecular factors [30,43,53,54]. Deviation in the programming of gene expression of PGC and sex-determining factors in fish by artificial means can lead to either sex reversal or sterility [17,19,55,56,57]. In the present study, we measured the basal mRNA expression of some important genes involved in the development of PGCs and SD in *S. salar* by qPCR.

The high sensitivity, reproducibility, precise quantification, fast readout and easy operation make qPCR one of the most popular and commonly used tools for measuring gene expression analysis in molecular biology [58,59,60]. However, the output results of qPCR are affected by many factors such as RNA quality and quantity, reverse transcription, HKGs and PCR efficiencies [61,62], meaning careful attention must be paid during nucleic acid isolation from the tissue. Good-quality RNA is essential for qPCR analysis because it serves as the starting material for reverse transcription, which generates cDNA templates for qPCR amplification [63]. RNA degradation, fragmentation, or contamination can lead to inaccurate results and unreliable gene expression analysis. Techniques such as RNA quantification, quality assessment, and sample preparation are important for obtaining high-quality RNA for qPCR analysis.

As discussed in the introductory section, RNA extraction from salmon eggs is very difficult, and in this study, we isolated the embryonic cells by dechorionation. The immersion in acetic acid made the eggs transparent, facilitating easy cell collection, while centrifugation in PBS did not alter cellular physiology. Agarose gel electrophoresis results demonstrated the two bands, i.e., 28s rRNA and 18s rRNA, which is an indicator of high-quality and intact RNA [64]. The RIN quantifies the fragmentation of the RNA sample [65], and high-quality RNA should fall in the range of a value of eight. In the present study, the RIN value was more than eight, which indicates that the isolated RNA had good integrity.

Housekeeping genes are constitutively expressed and assumed to have stable expression levels across different samples and experimental conditions. However, in reality, the expression of housekeeping genes may vary depending on the tissue type, developmental stage or experimental treatments. Therefore, it is important to validate the expression stability of housekeeping genes under specific experimental conditions before using them for normalization. This can be achieved by using statistical algorithms such as geNorm, NormFinder, BestKeeper etc. The delta-CT method used in the present study compares ΔCt of the HKGs in pairs, thereby bypassing the need for accurate RNA quantification [46]. The average SD index is used to rank the housekeeping genes based on their stability in this method. BestKeeper, which is an Excel-based statistical algorithm, uses pairwise correlations to determine the most suitable internal control gene [47]. NormFinder uses a model-based approach, provides a distinct analysis of the sample subgroups and includes both intra and inter-group variation estimations to calculate a gene stability value [48,66]. geNorm calculates the average expression stability value (M), a pairwise variation between one HKG and other candidate reference genes. During analysis, it excludes the gene with a high M value in each step, and the process is repeated until the most stable gene (least M value) is selected. GeNorm estimates pairwise variation (Vn/n + 1) to determine the optimal number of candidate reference genes, and they should satisfy Vn/Vn + 1 below the threshold value of 0.15 [49]. Finally, the comprehensive stability was estimated on the basis of output results from other validation algorithms as discussed in the data analysis section to select the HKG as an internal control.

In the present study, *UB2L3* was found to be the most stable HKG in the developing embryos of *S. salar* as per all the algorithms. Most of the ontogenetic studies on *S. salar* used the elongation factor gene [27,67,68] or 18s rRNA [69,70] as an internal control for qPCR studies. Additionally, in various gene expression studies of *S. salar*, the reference genes behave differently, as in immunity-related studies, where *β-actin* was stable [71,72,73,74] while the internal controls changed based on the stress factor (temperature stress, sea lice etc.) in an experiment [75,76,77,78]. So, the present results on HKG validation support the statement that a researcher should not rely on the so-called “golden standard” term and assume that the well-known housekeeping gene will work for every gene expression study [29].

The *vasa* transcript in the present study was expressed in unfertilized eggs and almost in all stages analyzed, but a significant decreasing trend was observed after 80 d°C. Moreover, after 430 d°C, the mRNA expression was significantly higher. Similarly, the *dnd1* transcript was expressed in unfertilized eggs until 30 d°C and was either negligible or absent in the remaining stages. The semi-quantitative PCR results in *S. salar* revealed that the *vasa* transcript was expressed up to the 10-somite stage, and *dnd1* was expressed up to the early blastula stage, which corresponds to 162 d°C and 30 d°C [at 6 °C], respectively [79]. In summer flounder (*Paralichthys dentatus*) and Nile tilapia (*Oreochromis niloticus*), both the transcripts were expressed up to the blastula stage [30,80]. The mRNA expression pattern of the two transcripts suggests that they are maternally inherited [79,81] and play an important role in the PGC specification and development. The sudden drop in mRNA expression between the blastula and gastrula stage is due to the degradation of the transcripts through an unknown mechanism, although some studies suggest that microRNAs are involved in degrading the maternal mRNAs [82,83,84]. Additionally, the genes could have been methylated because they have no role in certain stages [85]. In our study, the increase in *vasa* expression at 430 d°C, which corresponds to the yolk sac larval stage in *S. salar*, is in contrast with the results of the other teleost, in which low expression was detected [86,87,88]. In situ hybridization and immunohistochemistry detected the presence of *vasa* at a higher level in hatched larvae, which corroborates with the present results [79].

In the present study, the *S. salar nanos3a* was detected in the unfertilized egg, which suggests its maternal inheritance. The transcript began being expressed at 30 d°C and was drastically downregulated after 50 d°C, which suggests its role in PGCs for a shorter duration. This could indicate a role in migration or maintenance, but it needs to be further evaluated. Like our results, *nanos3a* expression declined in the later stages and was almost absent in the hatching stage in Atlantic cod (*Gadus morhua*) [89]. Similarly, *nanos3a* expression declined in zebrafish after 5 days post-fertilization [35].

In the present study, *sdf1* was expressed in all embryonic developmental stages except 80–210 d°C, which is in accordance with the expression results in medaka [40]. The expression of this transcript is high after hatching, which suggests its role in the later stages of development of PGCs or with other somatic cells such as pigment cells or muscle cells [90,91].

A network of genes is involved during embryonic stages to guide the bipotential gonad to change into either the testis or ovary [92]. The most common SD genes identified in teleosts include *gsdf*, *amh*, *cyp19a*1a, *foxl2* and *dmrt* variants [93,94]. *gsdf* and *amh* belong to the TGF-β superfamily of genes and control cell proliferation [95] and are mainly involved in male SD [96]. *gsdf* interacts with other male-specific genes such as sex dmy or *dmrt1*, and it affects estradiol production, a major female-specific hormone [97,98]. Its role in the regulation of PGC proliferation and germ cell differentiation has also been validated [99,100]. The role of *amh* in fish is to regulate male germ cell accumulation and prevent female-biased sex ratios [101]. Our results showed that *gsdf* and *amh* are weakly expressed during the embryonic development stages, and the mRNA expression was greater after hatching. In rainbow trout (Oncorhynchus mykiss), the RT-PCR results show that *gsdf* started being expressed at 2.5 dpf, and in zebrafish, the transcript was expressed after 16 dpf [99,102]. Our results of *gsdf* suggest that it is involved in later stages of PGC development in *S. salar*, which is supported by the fact that this transcript is restricted to somatic cells of the genital ridge surrounding PGCs in rainbow trout, and in *gsdf* morphants, the PGC proliferation was suppressed [99]. Similarly, *amh* expression is initiated in somatic cells when the PGCs are migrating in the somatic gonadal precursors [103,104]; therefore, it may play an important role in the maintenance of PGC in later developmental stages.

*cyp19a*1 is a member of the aromatase family of genes that are involved in the conversion of androgens to estrogens [105], and blocking its activity by aromatase inhibitors or gene editing results in a female-to-male sex reversal [106,107]. Like *gsdf* and *amh*, *cyp19a*1 was not expressed much in the embryonic stages in the present study, and the transcript started being expressed in hatched larvae, which is in consistent with the earlier reports stating its late-stage expression in fish [108,109]. The results indicate an unimportant role in the maintenance of PGC and its involvement mostly in the SD in fish [110].

*dmrt1* belongs to the family of genes characterized by having a highly conservative zinc-finger DNA-binding motif [DM domain], and it plays an important role in male SD [111,112]. On the other hand, *foxl2* is a member of the Fox gene family and is involved in ovarian differentiation and oogenesis in females [113]. Mutations in *dmrt1* resulted in male-to-female sex reversal, and the opposite is true for *fox12* mutations [111,114]. In the present study, *dmrt1* mRNA was expressed in all stages of embryonic development as well in the yolk sac larvae. In Yellow catfish (*Pelteobagrus fulvidraco*), *dmrt1* is expressed in all embryonic stages [115], while in ayu (*Plecoglossus altivelis*), it was expressed from day two post-fertilization [116]. *dmrt1* has been found to be expressed in the somatic cells surrounding PGC, hence playing a crucial role in germ cell development [117], which may be the reason for its wide expression in embryonic developmental stages. *foxl2* mRNA expression in the present study was weakly expressed up to 80 d°C and was increased from 110 d°C. In Celebes medaka (*Oryzias celebensis*), the *foxl2* transcript started being expressed from the late blastula stage and increased in the subsequent stages [118]. The available data on the role of *foxl2* in PGCs is limited, and the present study suggests that it may have a function prior to the onset of sexual differentiation. However, further investigation is required to fully understand its role in PGC development.

## 5. Conclusions

To summarize, the study successfully extracted high-quality RNA from developing *S. salar* embryos, which was then used to analyze the expression of PGC genes during embryogenesis. The results provide insights into the expression patterns of key PGC transcripts during early development and could serve as a reference for future studies on gene editing efficiency in *S. salar*.

## Figures and Tables

**Figure 1 animals-13-01094-f001:**
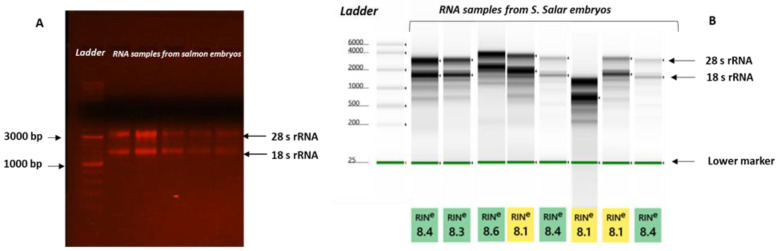
Gel electrophoresis of RNA (**A**). It showed clear distinctive bands of 28s rRNA and 18s rRNA. RNA analysis was carried out using the Agilent 2200 TapeStation system with a gel image showing different RNA samples (**B**).

**Figure 2 animals-13-01094-f002:**
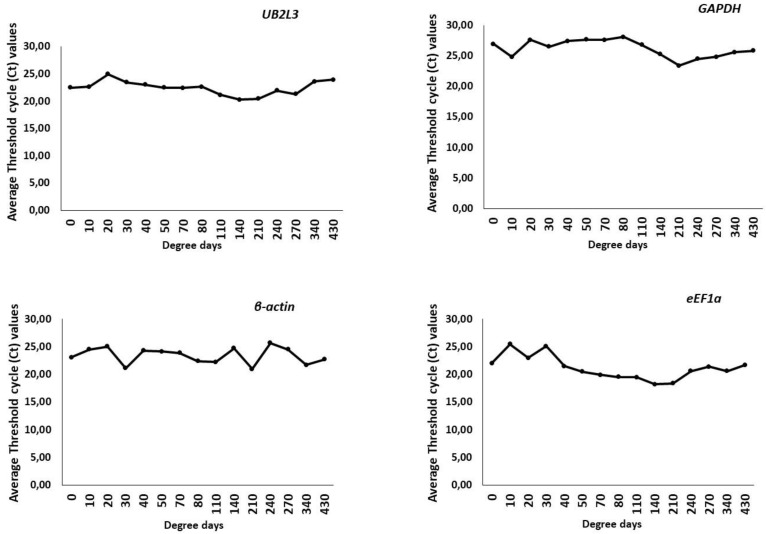
Average threshold cycle (Ct) values of different HKGs in the developmental stages of *S. salar*.

**Figure 3 animals-13-01094-f003:**
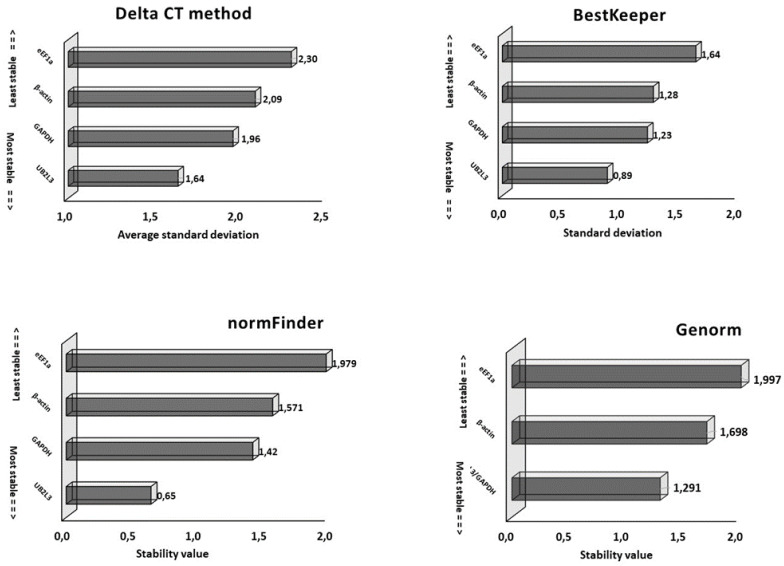
Evaluation of candidate reference genes using Delta Ct, BestKeeper, normFinder and Genorm statistical algorithms in different developmental samples of *S. salar.* The lower value indicates greater stability of the reference gene and vice versa.

**Figure 4 animals-13-01094-f004:**
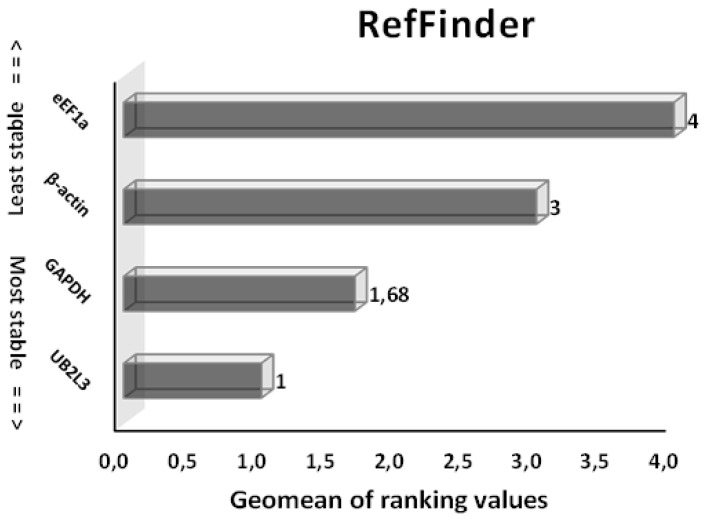
Stability ranking of candidate reference genes as determined by RefFinder in different developmental stages of *S. salar*. Lower Geomean values indicate more stable expression and vice versa.

**Figure 5 animals-13-01094-f005:**
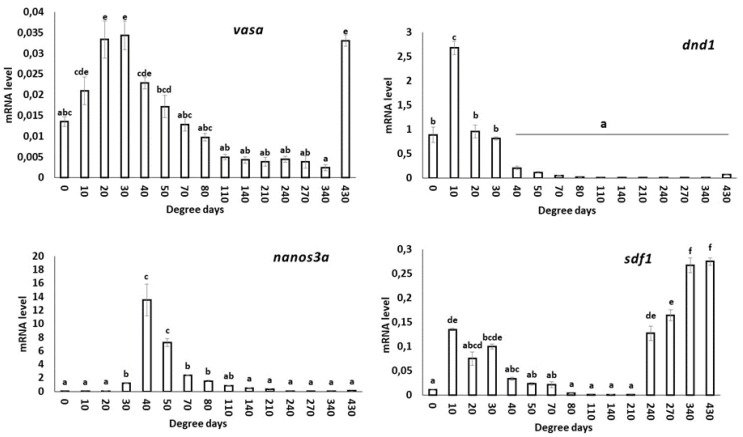
Basal mRNA transcript level of PGC genes: *vasa, dnd1, nanos3a* and *sdf1* during developmental stages of *S. salar*. Data are presented as mean ± SD (*n* = 4). Bars with different superscripts denote statistically significant differences (*p* < 0.05) between developmental stages. 0: unfertilized egg and 430: yolk sac fry or alevins.

**Figure 6 animals-13-01094-f006:**
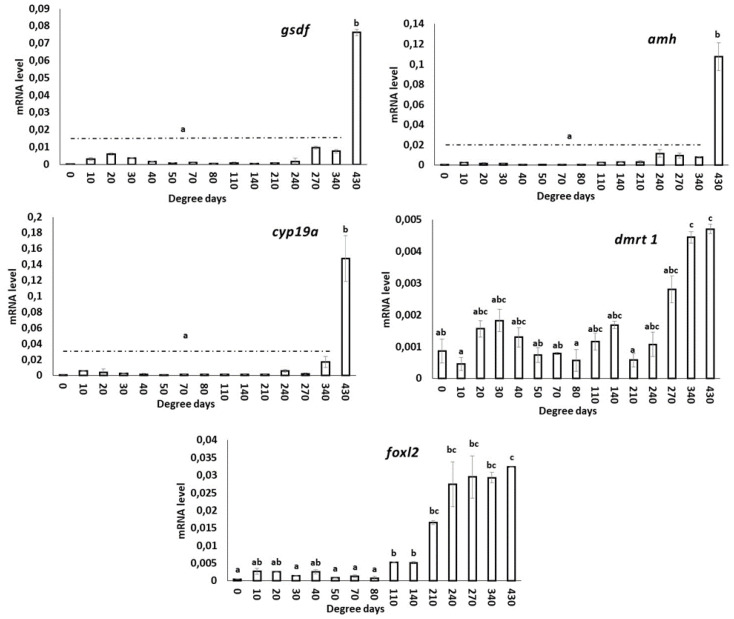
Basal mRNA transcript level of SD genes: *gsdf*, *amh*, *cyp19a*, *dmrt1* and *foxl2* during developmental stages of A. salmon. Data are presented as mean ± SD (*n* = 4). Bars with different superscripts denote statistically significant differences (*p* < 0.05) between developmental stages. 0: unfertilized egg and 430: yolk sac fry or alevins.

**Table 1 animals-13-01094-t001:** Incubation time of *A. salmon* eggs/embryos and larvae and the number of samples used in the present study. (Temperature was maintained at 10 °C).

Day\Days Post-Fertilization	Degree Days (d°C)	Pooled Number of Eggs/Embryos/Larvae per Sample
0	unfertilized egg	10
1	10	10
2	20	10
3	30	10
4	40	10
5	50	6
7	70	6
8	80	6
11	110	6
14	140	6
21	210	6
24	240	6
27	270	6
34	340	6
43	430 (yolk sac fry or alevins)	3

**Table 2 animals-13-01094-t002:** Primers used in the present study.

Gene	Forward Primer Sequence (5′ to 3′)	Reverse Primer Sequence (5′ to 3′)	Accession Number and Reference
*vasa*	CGCTCCCTGGTCAAAGTCCTGTC	GCTAGTTGACTCGCCCCATCTCTC	JN712912
*dnd1*	TCTGTACAGGGCCTGATGGT	TAAAACAAAGTAGGGGATCTGTG	[19].
*nanos3a*	ATGGAGTCCGAAAACAAGAGT	CGGTTCTGGGGTGAACTTGC	KC237283
*sdf1*	GTGTTGGTCCTACTGGCTGTGGC	GAGGGACGGTGTTGAGAGTGGAGC	NM_001140787
*gsdf*	GACAAAGCAGTGGCTGTACC	GGCAGCATTTCAGACCACTA	XM_014138924
*amh*	CAGTCACTCTCTGCAGCCTTACAA	CAACATTGAATCTCCATTTCAGTTTAC	NM_001123585
*cyp19a*	TCAAACAGAACCCTGACGTAG	GCTCCCTTTCACCTATAGCAGTGT	AF436885
*dmrt1*	AGGAGGAGGAGATGGGGCTCTGTA	CCAGCAGAGGTGTTTCCACAGGTAG	XM_014172771
*foxl2*	GCGGTGATGGGTACGGCTACCTG	GACGGGACTCACGTTGCCACTGG	JX184084
*β-actin*	CCAAAGCCAACAGGGAGAAG	AGGGACAACACTGCCTGGAT	BG933897
*GAPDH*	AAGTGAAGCAGGAGGGTGGAA	CAGCCTCACCCCATTTGATG	AM230811
*eEf1a*	CCCCTCCAGGACGTTTACAAA	CACACGGCCCACAGGTACA	BT058711
*UB2L3*	CGAGAAGGGACAGGTGTGTC	ACCAACGCAATCAGGGACT	NM_001141284

**Table 3 animals-13-01094-t003:** Statistical analysis of housekeeping genes by BestKeeper.

	*UB2L3*	*eEF1a*	*β-actin*	*GAPDH*
n *	15	15	15	15
geo Mean [CP]	21.90	20.57	22.88	25.56
AR Mean [CP]	21.93	20.67	22.93	25.60
min [CP]	20.00	18.00	20.00	23.00
max [CP]	24.00	25.00	25.00	28.00
std dev [±CP]	0.89	1.64	1.28	1.23
CV [% CP]	4.05	7.96	5.58	4.79
min [x-fold]	−3.74	−5.92	−7.38	−5.90
max [x-fold]	4.27	21.61	4.33	5.42
std dev [±x-fold]	1.85	3.13	2.43	2.34

* number of stages from which mRNA expression of HKGs was analyzed.

## Data Availability

The data presented in this study are available on request from the corresponding author.

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
