# Peer review of "Insights into Early Ontogenesis of Salmo salar: RNA Extraction, Housekeeping Gene Validation and Transcriptional Expression of Important Primordial Germ Cell and Sex-Determination Genes"

_animals, 2023, doi:10.3390/ani13061094_

Round 1

Reviewer 1 Report

Well done study and write up. The only thing I would suggest is removal of the intermediate lines in the Tables. 

Author Response

Reply: Thank you for the compliment. The suggestions have been incorporated

Reviewer 2 Report

This is a well-performed and interesting study. A good method for RNA extraction and quantification in eggs is described and used to provide new data on expression profiles of relevant PGC and SD genes during early development in Salmo salar. Data has both scientific and technical interest. I have a methodological concern. In section 2.2 of methods, authors mention that “Ten eggs/embryos until 40-degree days (d°C), six embryos 135 up to 340 d°C, and three yolk sac fry or alevins (only the trunk part was collected) were 136 collected and pooled together and four such pools were used to study the gene expression”. This means, in practice, that only 1 quadruplicated sample is used for analysis, reducing the solidity of data. Why the authors chose this option instead of analyzing individual samples?; it is rare to see that you collect 6 or 3 individuals, pool them and then analyze the pool as 4 “samples”. Please, clarify this point. Otherwise, the study is well written and presented and I would only have some minor comments to this ms. 

Minor comments

Methods (l. 118): it is mentioned “…..after being anesthetized with Aquacen Formaldehyde….”. This is not an anesthetic. Please, check and replace.

Methods, section 2.1: should provide more details of the artificial fertilization and egg incubation protocols, and methods to discriminate fertilized from unfertilized eggs. Check grammar of the sentence “The development of fertilized eggs was seen faster without producing any deformities”.

Author Response

Reply: We would like to express our gratitude to the reviewer for conducting a comprehensive evaluation of our manuscript. With regard to the pooling of samples, we resorted to this approach as the cell count at the early stages was insufficient for the isolation of RNA from individual eggs. Pooling is a commonly employed technique particularly when analyzing the basal expression of genes, as it effectively mitigates variability and reduces experimental expenses.

Minor comments

Methods (l. 118): it is mentioned “…..after being anesthetized with Aquacen Formaldehyde….”. This is not an anesthetic. Please, check and replace.

Methods, section 2.1: should provide more details of the artificial fertilization and egg incubation protocols, and methods to discriminate fertilized from unfertilized eggs. Check grammar of the sentence “The development of fertilized eggs was seen faster without producing any deformities”.

Reply: All the corrections have been incorporated into the main text of the manuscript.

Reviewer 3 Report

The authors reported an improved method to extract total RNA from the early embryos of Atlantic salmon and a validation of the house keeper genes for real-time RT-PCR. The researchers in salmon or other fishes may be interested to this study. The results were clearly shown and were soundness. However, the novelty is moderate and may be acceptable after revision.

1) There were numbers of  typos or/and grammatic errors in the language. E.g. line 19, "use it from"; line 37, "SD genes and were expressed"; line 66, "changing environment and is low"; etc.

2) The abbreviations must be indicated with its full name when it is appeared at first, such as A. salmon, qPCR, cds, etc.

3) The genes usually are written in italic. However, the genes were written in both italic and normal forms in this manuscript.

4) The abstract did not indicate which was the best of the house keeper genes for qRT-PCR.

5) The introduction and discussion parts can be focused on their studies of the method for RNA extraction and the house keeper genes.

Author Response

Reviewer 3:

The authors reported an improved method to extract total RNA from the early embryos of Atlantic salmon and a validation of the house keeper genes for real-time RT-PCR. The researchers in salmon or other fishes may be interested to this study. The results were clearly shown and were soundness. However, the novelty is moderate and may be acceptable after revision.

1) There were numbers of  typos or/and grammatic errors in the language. E.g. line 19, "use it from"; line 37, "SD genes and were expressed"; line 66, "changing environment and is low"; etc.

Reply: The corrections have been incorporated.

2) The abbreviations must be indicated with its full name when it is appeared at first, such as A. salmon, qPCR, cds, etc.

Reply: The corrections have been incorporated.

3) The genes usually are written in italic. However, the genes were written in both italic and normal forms in this manuscript.

Reply: Done

4) The abstract did not indicate which was the best of the house keeper genes for qRT-PCR.

Reply: The correction has been incorporated.

5) The introduction and discussion parts can be focused on their studies of the method for RNA extraction and the house keeper genes.

Reply: Some more sentences haven been included.